# Application of a 21-Gene Recurrence Score in a Swiss Single-Center Breast Cancer Population: A Comparative Analysis of Treatment Administration before and after TAILORx

**DOI:** 10.3390/diagnostics14010097

**Published:** 2023-12-31

**Authors:** Elena Diana Chiru, Anton Oseledchyk, Andreas Schoetzau, Christian Kurzeder, Raphael Mosimann, Marcus Vetter, Cvetka Grašič Kuhar

**Affiliations:** 1Medical Oncology, Basel University Hospital, 4051 Basel, Switzerland; anton.oseledchyk@usb.ch (A.O.); marcus.vetter@ksbl.ch (M.V.); 2Center of Oncology and Hematology, Cantonal Hospital Baselland, 4410 Liestal, Switzerland; 3Department of Biomedicine, Basel University, 4051 Basel, Switzerland; info@eudox.ch; 4Breast Center, Basel University Hospital, 4051 Basel, Switzerland; christian.kurzeder@usb.ch; 5Faculty of Medicine, Basel Medical University, 4051 Basel, Switzerland; raphael.mosimann@stud.unibas.ch; 6Department of Gynecologic Oncology, Basel University Hospital, 4051 Basel, Switzerland; cgrasic@onko-i.si; 7Medical Oncology Department, Institute of Oncology Ljubljana, SI-1000 Ljubljana, Slovenia; 8Faculty of Medicine Ljubljana, Korytkova 2, SI-1000 Ljubljana, Slovenia

**Keywords:** oncotype, recurrence score, breast cancer, genomic risk, chemotherapy, genomic assay, Exact Sciences, oncotype RS, oncotype DX, clinical risk, changes in chemotherapy, TAILORx

## Abstract

In patients with hormone receptor positive, human epidermal receptor 2 negative (HR+/HER2−) negative breast cancer (BC), the TAILORx study showed the benefit of adding chemotherapy (CHT) to endocrine therapy (ET) in a subgroup of patients under 50 years with an intermediate Oncotype DX recurrence score (RS 11–25). The aim of the present study was to determine if the TAILORx findings, including the changes in the RS categories, impacted CHT use in the intermediate RS (11–25) group in daily practice, as well as to identify the main factors for CHT decisions. We conducted a retrospective study on 326 BC patients (59% node-negative), of which 165 had a BC diagnosis before TAILORx (Cohort A) and 161 after TAILORx publication (Cohort B). Changes in the RS categories led to shifts in patient population distribution, thereby leading to a 40% drop in the low RS (from 60% to 20%), which represented a doubling in the intermediate RS (from 30% to 60%) and an increase of 5% in the high RS (from 8–10% to 15%). The overall CHT recommendation and application did not differ significantly between cohort B when compared with A (19% vs. 22%, resp., *p* = 0.763). In the intermediate RS (11–25), CHT use decreased by 5%, while in the high-risk RS category (>25), there was an increase of 13%. The tumor board recommended CHT for 90% of the patients according to the new RS guidelines in cohort A and for 85% in cohort B. The decision for CHT recommendation was based on age (OR 0.93, 95% CI 0.08–0.97, *p* = 0.001), nodal stage (OR 4.77, 95% CI 2.03–11.22, *p* < 0.001), and RS categories (RS 11–25 vs. RS 0–10: OR 0.06 (95% CI 0.02–0.17), *p* < 0.001; RS > 26 vs. RS 11–25: OR 618.18 95% CI 91.64–4169.91, *p* < 0.001), but did not depend on the cohort. In conclusion, while the tumor board recommendation for CHT decreased in the intermediate RS category, there was an increase being reported in the high RS category, thus leading to overall minor changes in CHT application. As expected, among the younger women with intermediate RS and unfavorable histopathological factors, CHT use increased.

## 1. Introduction

Breast cancer (BC) remains the most frequent cause of cancer death among women worldwide with over 685,000 deaths each year, accounting for almost 7% of the global mortality among female oncologic patients [1]. Over 80% of BC-related fatalities are due to the development of therapy-resistant metastatic diseases [2].

In hormone receptor positive, human epidermal growth factor receptor 2 negative (HR+/HER2−) BC patients, the decisions for adjuvant chemotherapy relies traditionally on tumor size, lymph node involvement, tumor grade, histologic type, and the Ki-67 proliferation index. These features have become core drivers in the creation of prognostic algorithms (e.g., Adjuvant! Online) and indexes (e.g., the Nottingham Prognostic Index) [3]. Recently validated gene expression signatures were introduced into clinical practice to help guide the decision for adjuvant chemotherapy [4,5,6].

The Oncotype DX genomic test was first published in 2004, and it was implemented in the clinical setting shortly thereafter. Its creation was based on the documentation of 250 genes from 447 different tumor samples from the National Surgical Adjuvant Breast and Bowel Project (NSABP) B-20 study [7]. Sixteen genes were chosen based on statistical association with recurrence rate, and five genes were deemed responsible for housekeeping. When compared to other available tests, Oncotype DX relies mostly (60%) on the estrogen module [8,9].

The RS range was set between a value of 0 and 100, with higher scores representing a higher risk for disease recurrence. The RS categories were initially defined by the manufacturer as low (<18), intermediate (18–30), and high (>31), as based on the results of the NSABP B-20 study [10].

Subsequently, they were validated in a retrospective analysis of the tumor samples stored in the NSABP Tissue Bank from patients treated with tamoxifen in the NSABP trial B-14, which was a large prospective U.S. study [11]. At 10 years, a 30.5% distant recurrence rate was found in the high-risk population (RS > 31), which is significantly higher when compared with patients in the low risk (RS < 18) and intermediate risk groups, which had a recurrence rate of 6.8% and 14.3%, respectively. Based on these findings, Oncotype DX and other signatures have come to support chemotherapy administration in patients with high genomic risk [7].

With the promotion and implementation of national screening programs across the U.S., which started in 2009, the incidence of early BC diagnosis, especially among HER+/HER2− tumors, increased significantly. These tumors are usually associated with a 5-year survival rate of >94% and a recurrence rate of 15%, and there is an unclear benefit to CHT [12,13]. Data analysis from the Surveillance, Epidemiology and End Results (SEER) on the impact of Oncotype DX testing on these tumors, however, showed an inverse correlation between CHT administration and RS testing. Genomic assay testing increased between 2004–2015 from 1.5% to 34%, thus leading to a 6% drop in CHT applications (from 42% to 36%) among the patients who underwent testing [14]. Moreover, patients who tested for RS also tended to have improved survival when compared with those who did not, even after adjustment for clinical variables. Specific analysis indicated a reduction of approximately 80% in breast cancer-specific risk of death among those patients who underwent testing. Consistent with RS testing recommendations [15], the use of adjuvant CHT increased among patients in the high RS groups and decreased in the intermediate and low risk groups. This analysis confirmed the implementation of Oncotype DX testing as a risk stratification tool, especially for node negative tumors [16,17].

Since its introduction to clinical practice, the ASCO [18], the National Comprehensive Cancer Network (NCCN) [15], the European Society for Medical Oncology (ESMO) [16], the National Institute for Health and Care Excellence [19], and the St. Gallen Consensus Conference [20] have included Oncotype DX testing into their breast cancer guidelines. In the U.S., Medicare started funding Oncotype DX screening in 2006 [21]. Ever since, this genomic assay has been validated through multiple prospective–retrospective, prospective non-randomized, and randomized studies [22].

One of the most relevant trials assessing the benefit of CHT among specific subgroups within the predefined RS categories was the Trial Assigning Individualized Options for Treatment (TAILORx) study. In the low RS (0–10) cohort of TAILORx, a lack of benefit of CHT when added to endocrine therapy was already established [23], while in the high RS (>25) category, the addition of CHT to ET was clearly associated with a longer 5-year disease-free survival (DFS) of 93%, which is longer than expected when using endocrine therapy alone (78.5% in NSABP B20 trial) [17].

Published in July 2018, TAILORx was a non-inferiority, prospective study on HR+/HER2− node negative patients, whereby it aimed to detail the benefits of CHT in intermediate RS 11–25 [24].

In all patients (i.e., the intention to treat population), the study demonstrated no benefit of CHT in the intermediate RS category (RS 11–25) (*n* = 6,711)—in other words, endocrine therapy (ET) was non-inferior to chemo-endocrine therapy (CHT-ET) in terms of invasive disease-free survival (IDFS) (HR: 1.08 [95% CI: 0.94–1.24]; *p* = 0.26). However, a subgroup analysis emphasized a clear benefit of CHT-ET vs. ET alone among premenopausal women under 50 years of age, with the added IDFS benefit at 9 years f 3.5% in the RS 11–15 subgroup, 9% in the RS 16–20 subgroup, and 6.3% in the RS 21–25 subgroup [24].

In this study, we looked at how the data derived from TAILORx analysis affected the daily clinical therapy decision-making process in daily clinical practice in a single oncology center from Basel, Switzerland to determine if any significant changes in CHT application occurred especially in the node-negative BC patient cohort. A secondary endpoint was to determine the leading therapy decision factors for CHT. We also looked at how the tumor board (TB) decision deviated from standard recommendation as based on RS results. The changes we expected to observe were mainly based on the lowering of RS thresholds and therefore on the changes in patient number.

This analysis was intended as an evaluation of the current level of genomic testing implementation in a Swiss BC cohort, as well as an assessment of the potential biased decision-making factors, in order to provide Swiss oncologists with relevant information to be able to recommend the expansion of Oncotype RS testing, the design of personalized medical solutions for their BC patients, and to avoid over-/undertreatment per the recommended guidelines in place.

## 2. Methods

### 2.1. Study Design

This is a retrospective cohort analysis of 326 HR+/HER2− BC patients, who were treated at Basel University Hospital and Cantonal Hospital Baselland from 2010–2021. The study was approved by the local ethics committee (Ethics Committee Nord-West-Schweiz, www.eknz.ch). Patient data are anonymized and collected in a password-protected file.

### 2.2. Inclusion Criteria

BC patients over 18 years of age with HR+/HER2− that had node-positive or node-negative tumors diagnosed between 2010 and 2021, with a valid RS score on file, and who underwent at least one form of therapy (surgery, ET, radiotherapy [RT] and/or CHT-ET) were eligible for this study. Patients with a metastatic disease were not eligible.

### 2.3. Objectives

The primary endpoint was to assess any change in therapy (CHT-ET vs. ET alone) in the BC population before July 2018 (Cohort A) and after July 2018 (the timepoint of the publication of TAILORx trial findings) (Cohort B) when adjusted for RS category thresholds as defined by the manufacturer and as modified in the TAILORx study protocol.

The secondary endpoint was to determine any change in the therapy administration in the intermediate RS, as adjusted to the TAILORx criteria for the intermediate RS category.

Lastly, we looked at the main factors affecting therapy decision making for BC patients in our centers to determine how Oncotype DX RS influenced TB decisions and therapy implementations.

Based on the TAILORx study findings, our first hypothesis was that CHT use will decrease in those with an RS of < 26, specifically in patients who are >50 years. The secondary hypothesis proposed that CHT recommendation will increase in patients who are <50 years with an RS of 16–25. Our hypothesis testing was founded by the assumption that, along with clinicopathological factors and RS results, the findings from the TAILORx study significantly affected decisions on CHT use.

### 2.4. Statistical Analysis

Numerical variables are presented as the means and standard deviation (SD), or the medians with quartile 1 (Q1) and quartile 3 (Q3). Categorical variables are presented as frequencies and percentages. Descriptive statistics were used to summarize the patient characteristics in the two cohorts and, where the sample size allowed, *T*-tests for independent distribution analyses were conducted. A non-parametric Wilcoxon rank sum test was performed for hypothesis testing the target parameters in the two cohorts. Statistics were based on 95% confidence intervals (CI), and hypothesis testing was made at a two-sided alpha value of 0.05.

Chi square tests were used to check for associations between various demographic and histopathological characteristics and CHT-ET use. Point biserial correlation analysis was performed to check for correlations between CHT-ET, RS result, and age in Cohort A and B. Multivariate logistic regression analyses were made to test for the impact of age, nodal status, tumor grade, tumor size, RS score category, and Ki-67% on CHT-ET treatment in Cohorts A and B, as well as in the entire study cohort.

## 3. Results

### 3.1. Patients’ Characteristics

Cohort A included 165 and Cohort B 161 patients, with a mean RS of 17.72 (SD 9.59) and 17.89 (SD 9.53) (*p* = 0.87), respectively. The demographics and tumor characteristics are presented in Table 1. The patients’ median age was similar in both cohorts (59 [Q1 51, Q3 67] in A and 58 [Q1 48, Q3 67] years in B); however, Cohort B had a significantly higher percentage of patients that were younger than 50 years when compared to Cohort A (34% vs. 24%, *p* < 0.01), which means that the Oncotype test was performed to possibly omit chemotherapy. Significantly more patients in Cohort A were overweight or had obesity (55%) vs. Cohort B (40%) (*p* < 0.001). Patients in Cohort A also had slightly more comorbidities (39% vs. 45%), but the difference was not statistically significant (*p* = 0.719; Table 1).

### 3.2. Tumor Characteristics

Patients in Cohort A and B were different regarding the percentage of tumors with high Ki-67, which was defined as Ki-67 ≥ 20%. Cohort B had a higher percentage of high Ki-67 tumors than Cohort A (39% vs. 32%, *p* = 0.010; Table 1). Cohort A presented with a higher percentage of pT1 tumors (55% in A vs. 51% in B), although this was not significant (*p* = 0.927), and a higher percentage of the ≥pN2 stage (4% in A vs. 1% in B); however, no significant differences in pN were found (*p* = 0.546). Both cohorts had the same percentage of node-negative patients (59%) and similar other characteristics. Briefly, three-quarters of the tumors were of a non-special type (NST) histology, and more than half were of grade 2 (54% in A and 57% in B, *p* = 0.811) with a mean Ki-67% expression of 19% (SD 0.009) in A and 20% in B (SD 0.11) (*p* = 0.844). The TNM stage was similar in both cohorts. Interestingly, 35% and 32% of the patients (in Cohort A vs. Cohort B, respectively) were of stage IA. Over 60% of the patients in both cohorts had a stage II tumor, whereas 6% in A and 8% in B had stage III tumors (*p* = 0.602).

### 3.3. Recurrence Score Results

The mean RS was similar in both cohorts (17.72 (SD 9.59, IQR 10) in Cohort A vs. 17.89 (SD 9.53, IQR 12) in Cohort B, *p* = 0.533). There were no significant differences in the RS distribution between the two cohorts when compared and when based on the manufacturer’s thresholds (*p* = 0.15 for low RS 0–18, *p* = 0.833 for intermediate RS 19–30 and *p* = 0.15 for high RS > 31), or when based on TAILORx thresholds for RS categories (*p* = 0.817 for low RS 0–10, *p* = 0.199 for intermediate RS 11–25 and *p* = 0.795 for high RS > 26), as illustrated in Figure 1. Importantly, Figure 1 shows a shift in the risk groups from low to intermediate when new thresholds, as based on TAILORx study results, were applied. According to the manufacturer’s recommendations, up to 60% of patients had low RS (RS < 18), while only around 20% had a low RS (according to TAILORx thresholds (RS < 11)). Conversely, according to the original manufacturer’s guidelines, 8–10% were previously high risk (RS > 30), but more (around 15%) were high risk (RS > 25) (according to TAILORx). Consequently, the number of patients in the intermediate risk category increased from 30% to over 60% when the new RS threshold (11–25) was applied.

### 3.4. Treatment Strategies

Most patients underwent breast conservative surgery (68% in A and 67% in B, *p* = 0.352) and adjuvant radiotherapy (RT) (78% vs. 76%, A vs. B, resp., *p* = 0.223), with a mean dose that was higher in A (54.99 Gy, SD 8.31) vs. B (52.35 Gy, SD 7.88) but was not significant (*p* = 0.417). More than half of the patients underwent reconstruction (60% in cohort A, and 55% in B, *p* = 0.463) (Appendix A). The treatment rates with ET and CHT-ET are presented in Figure 2 and Appendix A, respectively. To summarize, when comparing the percentage of patients treated with ET and CHT-ET pre- and post-TAILORx (i.e., in Cohort A vs. Cohort B), there was almost no difference. Around 70% of the patients were treated with ET alone, and 30% with CHT-ET. However, when comparing them according to the RS, it was obvious that no patient received CHT in the low risk group vs. 3% (post- vs. pre-TAILORx). CHT usage in the RS 11–25 and RS ≥26 groups increased by 1% and 13% post- vs. pre-TAILORx, respectively. CHT refusal was lower in Cohort B. Ninety-six percent of patients in Cohort A and 94% in Cohort B were treated with ET, the majority of them with aromatase inhibitors (65.4% in A vs. 52% in B, *p* = 0.113). Six patients (4%) in Cohort A and 9 (6%) in Cohort B refused ET.

### 3.5. Analysis of Chemoendocrine and Endocrine Therapy Use

First, we analyzed the differences in the CHT-ET use between the cohorts according to the RS groups 0–10, 11–20, 21–25, 21–25, 26–20, and >30 in order to see which RS intervals were the most affected by the change brought about by TAILORx (Table 2). The greatest change for less CHT use was in the RS 21–25 group (from 35% to 17%). We also analyzed the impact of other characteristics on CHT-ET use. In Cohort B, we observed an increase in the use of CHT in patients of <50 years by 12.5%, in lobular carcinoma by 10%, and in node-negative, node-positive and grade 3 tumors by 2–3% (Figure 3, Appendix A). The use of ET was high in both cohorts (96% in Cohort A and 95% in Cohort B). The difference according to the tumor and patient characteristics were small (-4% to +3%), except for grade 3 tumors, which is where ET use increased by 58% (Figure 4, Appendix A).

### 3.6. Implementation of RS Guidelines into TB Decision

The comparison of the TB decisions with RS guidelines post-TAILORx showed that—in Cohort A—90% (*n* = 46/51) of patients and—in B—85% (*n* = 41/48) of patients were assigned to CHT-ET. This speaks for a relatively modest reduction in TB recommendations for CHT-ET, i.e., of only 5%. However, since, in Cohort B, there are statistically more patients aged <50 years (54/161 vs. 40/165 in Cohort A, *p* < 0.001), decisions could have been affected by that. Consequently, the TB decision for ET alone increased from 70% in Cohort A to 73% in Cohort B (Appendix A).

### 3.7. Implementation of TB Decisions in Clinical Practice

The decision of the TB on optimal adjuvant systemic therapy (CHT-ET or ET) was not fully implemented. Implementation was numerically higher in Cohort B vs. A (83% vs. 75%), but this was not statistically significant (*p* = 0.522). However, the implementation of CHT-ET in Cohort B was significantly higher when compared to Cohort A (73% vs. 61%, *p* < 0.001). The recommendation for TB in ET was implemented in most cases (98% in A and 97% in B, *p* = 0.824). All of the relevant information pertaining to therapy administration in the two populations is presented in Appendix A.

### 3.8. Logistic Regression

In Cohort A, the decision for CHT administration was influenced by younger age (*p* = 0.03), intermediate and/or high RS result (*p* < 0.001), node-positive status (*p* = 0.003), and higher tumor stage (*p* = 0.043). In Cohort B, on the other hand, the decision for CHT-ET was influenced only by younger age (*p* < 0.001) and node-positive status (*p* = 0.013) (Table 3).

Finally, in the entire population, a logistic regression analysis of the factors affecting CHT-ET administration showed that the model was significant at a Chi^2^ (2) = 13, *p* < 0.001, and that the CHT-ET was significantly influenced by age (*p* = 0.001), pN status (*p* < 0.001), and Oncotype RS score in all categories (low 0–10, intermediate 11–25, high ≥26, *p* < 0.001), as shown in Table 4. Interestingly, it was not influenced by cohort.

## 4. Discussion

Our findings show a very modest reduction in the TB recommendation of CHT-ET from 31% before TAILORx to 30% in the post TAILORx era, with the greatest change for less CHT use in the RS 21–25 group (from 35% to 17%). With regard to the TB recommendation for ET, the percentages remained high but were not statistically significant (from 96% in A to 95% in Cohort B), while the application tended to decrease by 2%, from 96% in A to 94% in B. When looking at what determined CHT-ET application, according to the multivariate logistic regression analysis, RS group, nodal status, and age were significant in the decision-making process.

When selecting our data according to RS categories, we observed a 5% reduction in the CHT-ET recommendation in the intermediate RS category 11–25, wherein TB was recommended in 22% (*n* = 23) of patients in Cohort A and in 17% (*n* = 16) of patients in Cohort B. However, due to the high refusal rate in A (33%) vs. B (15%), CHT-ET was administered in 19% of the patients from Cohort A, and 22% in Cohort B, thus showing that there was no significant change in the treatment applications over time (*p* = 0.763).

While a reduction of 18% (from 35% in Cohort A to 17% in Cohort B) was noted in the intermediate RS interval 21–25 group, it was balanced by an increase of 7% (from 6% in Cohort A to 13% in Cohort B) in the CHT-ET applications among patients with intermediate RS 11–20, which is contrary to other study findings (like those reported by Tesch et al. [25]). However, an increase in CHT-ET was noted among the patients with a higher RS (>30), wherein CHT applications increased from 57% to 78%, which is a finding that is similar to others reported [25].

There was no relevant change in the CHT-ET administration to be reported according to ages of <50 years old (*p* = 0.59) or >51 years old (*p* = 0.066), although we noted an increase of 12.5% in the CHT-ET administration from A to B among patients under 50 years of age. Among older patients (age > 50 years), an equal percentage of CHT-ET (19%) was administered in A and in B, thus confirming the maintenance of older practices in this age group.

However, a younger age did not significantly affect CHT-ET application in A (*p* = 0.835), nor B (*p* = 0.552), but older age did have more of an impact (*p* = 0.009 in A and *p* < 0.001 in B, respectively) This should explain why older patients tended to receive ET more often, i.e., without CHT. The RS result was significant when deciding for CHT among younger women in A (*p* = 0.009) and in B (*p* = 0.001), which can explain the 12.5% increase in the CHT-ET applications among younger women.

Interestingly, our logistic regression analysis on the possible factors that might have influenced CHT-ET applications showed the significant impact of age as a continuous variable (*p* = 0.03 in A and *p* < 0.001 in B), as well as nodal status (*p* = 0.034 in A and *p* = 0.013 in B). Also, the physicians before TAILORx seemed to have been driven more by tumor size (*p* = 0.043), and, contrary to expectations, by the intermediate and high RS results (*p* < 0.001) than after TAILORx publication (*p* = 0.678).

In terms of other treatments, there were no differences in surgery (68% in A and 67% in B, *p* = 0.114), surgical breast reconstruction (60% in A, 55% in B, *p* = 0.679, ET (96% in A vs. 94% in B, *p* = 0.306), radiotherapy (78% in A and 76% in B, *p* = 0.223), and osteo-oncologic therapy (41% of patients in A and 45% in B, *p* = 0.695).

Our findings did show a slight reduction of 1% in the CHT-ET TB recommendation after TAILORx, but this was due to more patient compliance. Also, there was a broader implementation of TB recommendations from 76% in A to 83% in B (*p* = 0.522). However, the TB recommendation did not always seem to be aligned to the guidelines. In our Swiss cohort, about 10–15% of patients remained undertreated when based solely on TB decision. When accounting for patient refusal, the difference became even more poignant with a 61% concordance in CHT-ET administration and RS-based recommendation before TAILORx publication and 73% after TAILORx publication.

When compared to the population selected for the TAILORx study, the median age in our cohorts was 59 years old, which is slightly older than in TAILORx (55–28 years old) [25] due to there being 74% postmenopausal patients vs. TAILORx (where postmenopausal patients accounted for 64% to 71% [26]). The results are also comparable to those reported in a similar prospective study that was conducted in Brazil. Only 9% of our patients had grade 1 tumors vs. 26% in TAILORx, a finding that is similar to others reported in similar analyses. The average tumor size in our cohort was also larger than that reported in TAILORx (2.5 cm vs. 1.7 cm in TAILORx cohort), while 40% of patients had node-positive disease, which is similar to the cohort analyzed by Mattar et al., in which 32% of tumors were node-positive [25]. This may explain the relative maintenance of therapy practices before and after TAILORx publication; however, it also speaks against the decision to recommend less CHT-ET than otherwise recommended by RS-based guidelines.

Our results also contradict those reported in the Italian study conducted by Cognetti et al., which showed significant changes in the treatment recommendation for 1683 patients after physicians were presented with the RS result. As such, CHT-ET recommendation dropped by 51%, while hormone therapy increased by 35%. According to these findings, 12% of patients would have otherwise been undertreated, and 49% of patients assigned for CHT-ET would have been overtreated [27]. In our study, 10% of patients in A and 15% in B would have been, according to the TB decision, undertreated; however, these results were further inflated through patient refusal, so that the 39% patients in A and 33% in B were, in fact (according to RS guidelines), undertreated. One may argue that, since the results from TAILORx were not known prior to July 2018, CHT was not necessarily warranted among our 23 patients with an intermediate RS of 16–25. However, the trend was maintained and deepened even after the publication of TAILORx, when only 85% of patients that would have been eligible for CHT were actually assigned to CHT.

Our findings align with others reported in the literature [27]. With regard to the CHT treatment in the node-positive cohort (28% had CHT vs. 25–28% in our cohorts, respectively) but this differed in the CHT applications in older populations (32% in the Italian cohort vs. 19% in our study) [27].

Regarding CHT, undertreatment among older Swiss BC patients was maintained in the post TAILORx era, and this was likely due to an interplay of patient refusals and physician decisions. This finding supports the necessity to magnify the focus on geriatric oncology. Comprehensive geriatric assessment and the inclusion of older non-frail patients in clinical trials could open the way to assessing the benefit of CHT among this segment of the patient population.

Similar results were reported when CHT was analyzed in lobular BC (23% vs. 24% in our B cohort) and in women under 51 years old (32% vs. 30% in our B cohort). However, the results differed for older patients with much less CHT applications in our population (19% vs. 32% in the Italian cohort), grade 3 tumors (31% vs. 53% in results reported by Cognetti et al.), and Ki-67 (31% in our B cohort vs. 40% in the Italian study) [27].

Mattar et al. also reported a significant reduction of 66% in the CHT-ET recommendation at two hospitals in Brazil, and they compared it to other results reported in Europe (38% reduction) and Mexico (28% reduction). However, this prospective study was conducted by means of survey with six licensed oncologists before and after knowing the RS results. Per the design itself, physician decisions might have been biased [26].

On the contrary, due to being a retrospective analysis that looks at interdisciplinary TB decisions in a relatively heterogenous BC patient population, our analysis reflects more of the everyday clinical reality pertaining to Swiss patients and physicians.

There are several strengths to our study. Firstly, our institutions have extensive experience with Oncotype DX testing. Secondly, with over 320 patients treated at a single oncological center that was divided between two hospitals, our study represents a relatively homogenous population. Thirdly, the information retrieved from electronic database provided us with realistic information of what was actually happening in the daily clinical setting and at interdisciplinary tumor board meetings.

The limitation of our study stems mainly from its retrospective design and its short follow-up period, especially for Cohort B. Also, the population included in the study, while it might be representative for the region, might not reflect the demographics of the entire Swiss BC population. Additionally, only 326 patients underwent testing in 11 years, which might point to a bias in selection. A prospective trial exploring the decision making for BC patients before and after the sharing of RS results would help determine the BC population who derive the most benefit from genomic assay testing, and this is currently being planned in our institution.

Our study shows how often BC patients undergo genomic testing, how often therapy implementation is made based on RS-guidelines, and what are the reasons behind each therapy decision the Swiss oncologists make in the clinic. This knowledge can help physicians navigate through potential biases, as well as to design optimized decision-making instruments to assist in personalized solutions for BC patients, or to present their cases in a geriatric tumor board where applicable. Additionally, this information can help increase patient participation in decision-making processes, as well as could increase compliance and adherence to treatment.

Contrary to most of the findings reported in similar studies, the Swiss BC population does not seem to be plagued by overtreatment, which might also point to better financial practices among Swiss oncologists. While physicians at our center are still orienting treatment decision making according to nodal status, RS result, and age, the final treatment decision seems to be rather personalized. For study validation, further analysis of the outcomes, especially among those patients who refused the administration of CHT or who were spared from CHT administration by TB decision, is highly recommended and should be pursued in the future. Also, since the RxPONDER publication in December 2021 [27], an analysis of the treatments and outcomes of node-positive BC patients who underwent Oncotype RS testing should also be warranted to establish the role of menopausal status in treatment decision making and outcomes in a real-world clinical setting.

## 5. Conclusions

Our findings indicate that the post-TAILORx tumor board recommendation for CHT decreased in the intermediate RS category, most notably in the RS 21–25 group (from 35% to 17%), but there was an increase in the high RS category, thereby leading to overall minor changes in CHT application. As expected, the RS result, nodal stage, and age significantly contributed to the decision for CHT use. CHT-ET use remains relatively unchanged for older patients, while it increased by 12.5% among younger patients. Our cohort seems to be undertreated with regard to post-TAILORx recommendation, and patient refusals add to this trend.

## Figures and Tables

**Figure 1 diagnostics-14-00097-f001:**
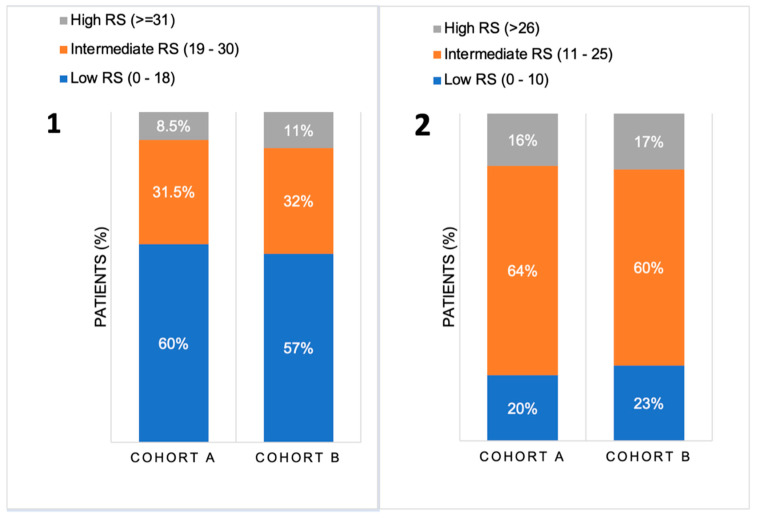
Distribution of the patients in Cohort A (before TAILORx) and Cohort B (after TAILORx) in each RS category (low, intermediate, and high) according to the different thresholds, as defined by the manufacturer recommendation (**part 1**) and by the TAILORx study design (**part 2**). RS = recurrence score. Note: percentages were rounded.

**Figure 2 diagnostics-14-00097-f002:**
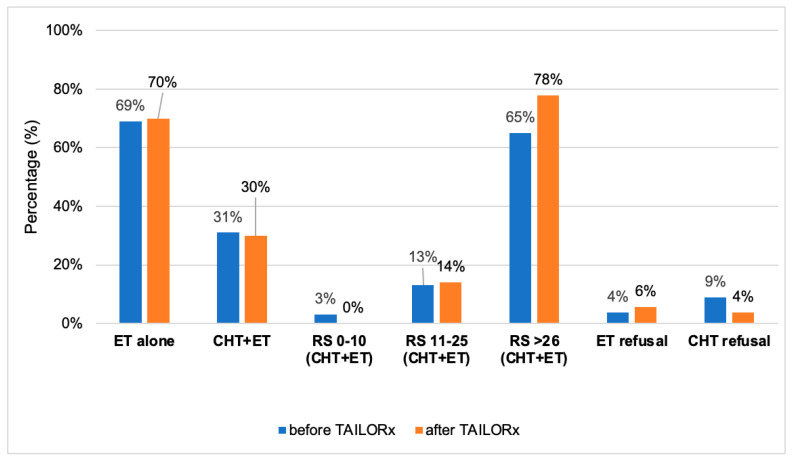
The percentage of patients treated with endocrine therapy (ET) and chemotherapy followed by endocrine therapy (CHT-ET), and the distribution of ET and CHT-ET according to the three RS groups, as determined by tumor board (TB) recommendation. In addition, the percentage of patients who refused ET or CHT + ET (from the entire population). ET = endocrine therapy; CHT = chemotherapy. Note: percentages were rounded.

**Figure 3 diagnostics-14-00097-f003:**
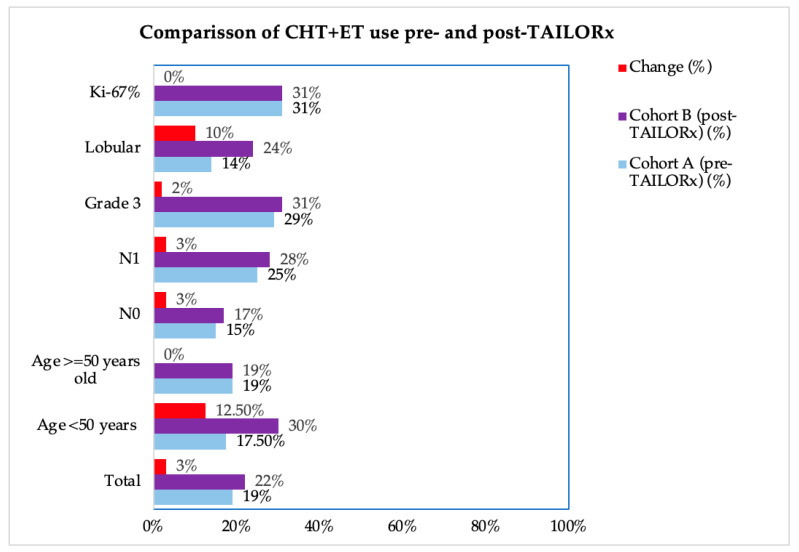
Change in the chemotherapy use in Cohort B (pre-TAILORx) and Cohort B (post-TAILORx) according to the patient and tumor characteristics. CHT-ET = chemoendocrine therapy. Note: percentages were rounded.

**Figure 4 diagnostics-14-00097-f004:**
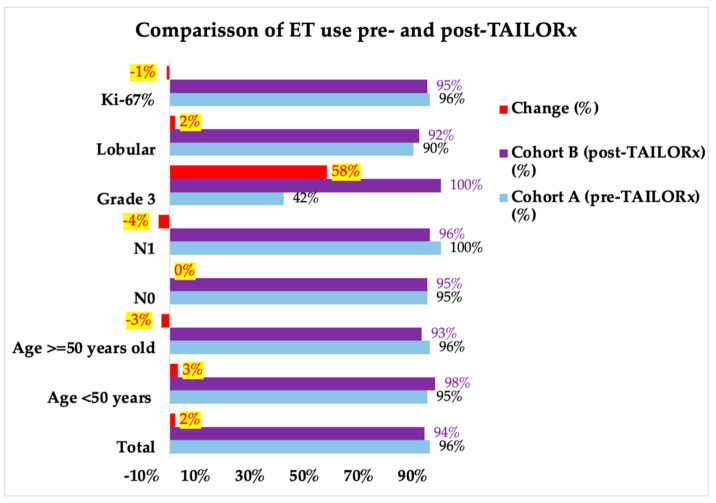
Comparison of endocrine therapy (ET) in patients in the two cohorts (pre- and post-TAILORx: Cohort A and Cohort B) and the observed changes in treatment. ET = endocrine therapy. Note: percentages were rounded.

**Table 1 diagnostics-14-00097-t001:** Demographic and tumor characteristics of patients evaluated with the Oncotype Dx test in the two cohorts, before (A) and after (B) publication of the TAILORx study.

	Cohort A *N* = 165	Cohort B *N* = 161	*p*-Value
Men *n* (%)	2 (1)	5 (3)	0.799
Age median (Q1, Q3)	59 (IQR 16, Q1 51, Q3 67)	58 (IQR 19, Q1 48, Q3 67)	0.937
Age ≤50/>50 years old *n* (%)	40 (24)/125 (76)	55 (34)/106 (66)	<0.001
Postmenopausal *n* (%)	122 (74)	119 (74)	0.922
Comorbidity *n* (%)	75 (45)	62 (39)	0.718
Multimorbidity *n* (%)	22 (13)	13 (8)	0.821
ASA score (median, IQR))	2 (1)	3 (1)	0.262
Body mass index (BMI) *n* (%)			
BMI < 18.5 kg/m^2^ *n* (%)	7 (4)	3 (2)	<0.001
BMI 18.5–24.9 kg/m^2^ *n* (%)	63 (40)	90 (58)	
BMI 25–29.9 kg/m^2^ *n* (%)	57 (37)	38 (25)	
BMI 30–34.9 kg/m^2^ *n* (%)	18 (12)	17 (11)	
BMI > 35 kg/m^2^ *n* (%)	11 (7)	6 (4)	
Histology			0.544
NST *n* (%)	125 (76)	125 (78)	
Lobular *n* (%)	30 (18)	26 (16)	
Mixed/others *n* (%)	6 (4)	10 (6)	
Grade			0.811
Grade 1 *n* (%)	11 (7)	18 (11)	
Grade 2 *n* (%)	85 (54)	90 (57)	
Grade 3 *n* (%)	61 (39)	49 (31)	
Tumor			
Size of tumor mean (SD)	25.26 (19.68)	24.78 (16.33)	0.573
pT1	93 (55)	82 (51)	0.927
pT2	56 (34)	64 (40)	
pT3 and over	16 (10)	15 (9)	
Nodal characteristics			
pN0 *n* (%)	98 (59)	95 (59)	0.546
pN1 *n* (%)	61 (37)	64 (40)	
pN2 or higher *n* (%) TNM	6 (4)	2 (1)	
Ki67 mean (SD)	0.19 (0.1)	0.2 (0.11)	0.844
Ki67 low (0–20%) *n* (%)	113 (68%)	98 (61%)	0.010
Ki67 high n (>20%) (%)	52 (32%)	63 (39%)	
Stage *n* (%)			0.602
IA	58 (35)	52 (32)	
IB	5 (3)	0 (0)	
IIA	59 (36)	62 (39)	
IIB	33 (20)	34 (21)	
IIIA	7 (4)	8 (5)	
IIIB	0 (0)	2 (1)	
IIIC	3 (2)	3 (2)	

TNM = tumor, node, metastasis stage; NST = non-specific type; SD = standard deviation; IQR = interquartile range; Q1 = quartile 1, 25%; and Q3 = quartile 3, 75%. Note: percentages were rounded.

**Table 2 diagnostics-14-00097-t002:** Chemotherapy (CHT) administration in Cohort A and Cohort B according to the recurrence score (RS) intervals.

	Cohort A		Cohort B		
RS Result	All *N* (%)	CHT *n* (%)	*N* (%)	CHT *n* (%)	*p*-Value
0–10	34 (21%)	1 (3%)	38 (24%)	0 (0%)	0.763
11–20	79 (48%)	5 (6%)	72 (45%)	9 (13%)	
21–25	26 (16%)	9 (35%)	24 (15%)	4 (17%)	
26–30	12 (7%)	9 (75%)	9 (6%)	7 (78%)	
>30	14 (8%)	8 (57%)	18 (11%)	14 (78%)	

RS = recurrence score; CHT = chemotherapy; and *N* = number. Note: percentages were rounded.

**Table 3 diagnostics-14-00097-t003:** Logistic regression model for the associations of patient, tumor, and treatment characteristics with chemoendocrine therapy in Cohort A (before TAILORx) and Cohort B (after TAILORx).

**Cohort A**		
	**OR (95% CI)**	***p*-Value**
Age	1.05 (1.01–1.11)	0.03
Age <50/>50	0	0.998
N0/+	3.32 (1.09–10.06)	0.034
Grade 3 vs. G1/2	0.4 (0.14–1.12)	0.082
Ki-67% high/low (<20% vs. ≥20%)	2.15 (0.74–6.29)	0.162
Tumor size (T2/3 vs. T1)	0.97 (0.95–1)	0.043
RS 18–30	0.12 (0.03–0.43)	0.001
RS > 31	0.04 (0.01–0.21)	<0.001
**Cohort B**		
	**OR (95% CI)**	***p*-Value**
Age	0.89 (0.84–0.94)	<0.001
Age <50/>50	0.79 (0.11–5.92)	0.818
N0/+	3.31 (1.29–8.53)	0.013
Grade 3 vs. G1/2	0.79 (0.27–2.26)	0.654
Ki-67 high/low (<20% vs. ≥20%)	1.1 (0.39–3.13)	0.853
Tumor size (T2/3 vs. T1)	0.99 (0.96–1.01)	0.357
RS 11–25	0.76 (0.21–2.76)	0.678
RS > 26	1.31 (0.36–4.76)	0.678

RS = recurrence score; N0/+ = nodal status; OR = odds ratio; and CI = confidence interval.

**Table 4 diagnostics-14-00097-t004:** Logistic regression analysis of the chemotherapy administration in the whole study population according to patient and tumor characteristics.

	OR (95% CI)	*p*-Value
Age	0.93 (0.89–0.97)	0.001
Age ≤50/>50 years old	1.35 (0.7–2.59)	0.367
Cohort A/Cohort B	0.47 (0.19–1.16)	0.102
Comorbidities Yes/No	1.24 (0.52–2.99)	0.626
pT	1.81 (0.92–3.56)	0.086
pN N0/+	4.77 (2.03–11-22)	<0.001
Lobular/ductal histology	0.61 (0.33–1.14)	0.12
Ki-67 ≤20%/>20%	0.6 (0.23–1.55)	0.291
Grade 3 vs. G1/2	2.45 (0.25–24.34)	0.444
RS intermediate (11–25) vs. low (0–10)	0.06 (0.02–0.18)	<0.001
RS high (≥26) vs. intermediate (11–25)	618.18 (91.64–4169.91)	<0.001

RS = recurrence score; pT = tumor stage; pN = nodal status; OR = odds ratio; CI = confidence interval.

## Data Availability

The data presented in this study are available on request from the corresponding author. The data are not publicly available due to data protection regulations of the participating medical institutions.

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
