# Peer review of "Application of a 21-Gene Recurrence Score in a Swiss Single-Center Breast Cancer Population: A Comparative Analysis of Treatment Administration before and after TAILORx"

_diagnostics, 2023, doi:10.3390/diagnostics14010097_

Round 1
Reviewer 1 Report
Comments and Suggestions for Authors
First of all, I would like to thank you for inviting me to review the manuscript entitled: 'Application of a 21-gene recurrence score in a Swiss single center breast cancer population. A comparative analysis of treatment administration before and after TAILORx’. The aim of the study was to estimate how data derived from TAILORx analysis affected the daily clinical therapy decision-making process in a single oncology center from Basel, Switzerland, in order to determine if any significant changes in CHT application occurred especially in the intermediate RS, node negative BC patient cohort. The general conclusion demonstrate maintenance of therapy practices and recommendations in Swiss cohort post TAILORx publication, with relative tendency for undertreatment especially when considering patients over 50 years of age. Thus, I recommend publication after some major issues have been addressed:
Major:
1. The introduction is too long, please focus on the main facts, which introduce to the research.
2. Lines from 94-135 sound like discussion.
3. Abstract should be modified, since it is chaotic; lines from 13-18 are not enough. Moreover, sentence: We conducted a retrospective study on 326 BC patients, of which 165 had a BC diagnosis before TAILORx (cohort A) and 161 after TAILORx publication (cohort B) and what next?
4. In line 397: Future implications: to analyse the performing of Oncotype in N+ patients, similar to Rx PONDER study…… what next?
5. Lines 421-423 should be removed, since in this part authors should provide their conclusion, but not discussion of their results.
6. Statistical analysis should be presented in a separate subsection and all statistical tests should be provided.
7. In Methods section, authors should provide subsections, including study design, inclusion and exclusion criteria, etc.
8. In respect to patients age (Table 1): 59 (16) and 58 (19). Authors indicated IQR, but they did not present Q1 and Q3 range only one value is provided: (16) or (19)?
9. Please, provide range for Ki67 (low) or (high) (Table 1).
10. Also, in discussion section please add paragraph related to clinical and practical aspects of the study.
11. How we can applicate your results into practice?, why your work is valuable in the field?
12. Conclusions should be shortened and the authors should clearly present a summary of the research.
Minor points:
1. Modification of the grammar and punctuation is required.
2. Please provide strength of the study (Line 384).
General: interesting, but chaotic work.
Moderate editing of English language required.
Author Response
Dear Ms. Peng,
Thank you for giving me the opportunity to submit a revised draft of my manuscript titled Application of a 21-gene recurrence score in a Swiss single center breast cancer population. A comparative analysis of treatment administration before and after TAILORx
to International Journal of Molecular Sciences.
We appreciate the time and effort that you and the reviewers have dedicated to providing your valuable feedback on my manuscript. We are grateful to the reviewers for their insightful comments on my paper.
We have been able to incorporate changes to reflect most of the suggestions provided by the reviewers. We have highlighted the changes within the manuscript. Here is a point-by-point response to the reviewers’ comments and concerns.
Comments from Reviewer 1
Comments and Suggestions for Authors
First of all, I would like to thank you for inviting me to review the manuscript entitled: 'Application of a 21-gene recurrence score in a Swiss single center breast cancer population. A comparative analysis of treatment administration before and after TAILORx’. The aim of the study was to estimate how data derived from TAILORx analysis affected the daily clinical therapy decision-making process in a single oncology center from Basel, Switzerland, in order to determine if any significant changes in CHT application occurred especially in the intermediate RS, node negative BC patient cohort. The general conclusion demonstrate maintenance of therapy practices and recommendations in Swiss cohort post TAILORx publication, with relative tendency for undertreatment especially when considering patients over 50 years of age. Thus, I recommend publication after some major issues have been addressed:
- 1. The introduction is too long, please focus on the main facts, which introduce to the research.
Response: Following your suggestion, we shortened the introduction significantly and included a paragraph to state the intention of our analysis.
Lines from 94-135 sound like discussion.
Response: In order to avoid discussion of TAILORx and subsequently derived analyses we decided to completely take out this part.
Abstract should be modified, since it is chaotic; lines from 13-18 are not enough. Moreover, sentence: We conducted a retrospective study on 326 BC patients, of which 165 had a BC diagnosis before TAILORx (cohort A) and 161 after TAILORx publication (cohort B) and what next?
Response: Thank you for this suggestion. After extensive editing and review of the statistical analysis results, the abstract was rewritten to include the main findings and conclusions.
In line 397: Future implications: to analyse the performing of Oncotype in N+ patients, similar to Rx PONDER study…… what next?
Response: In lines 982-988 we completed the paragraph by suggesting evaluation of results in view of the RxPONDER study results, to determine role of menopause on therapy decision making and outcome.
Other implications suggested throughout our discussion pertain to prospective exploration of the decision-making process before and after knowing the RS result, in order to identify possible biases in the selection of population to undergo testing (lines 948-951 of the clean manuscript version).
Lines 421-423 should be removed, since in this part authors should provide their conclusion, but not a discussion of their results.
Response: The lines were removed per your suggestion.
Statistical analysis should be presented in a separate subsection and all statistical tests should be provided.
Response: Statistical analysis methods and tests were included as a subsection in Methods (352-366 of the clean manuscript version). Additionally, we included Tables 3 and 4 to support our logistic regression analysis and in Figure 1 the two different sections were renamed “part 1” and “part 2” to avoid confusion of previously labeled “A” and “B” with cohorts A and B respectively.
In Table 1 an editing error was identified and changed, as 1 Patient with low RS received chemotherapy in cohort A and not in cohort B as previously mentioned.
In Figure 4 the change in endocrine therapy administration in grade 3 tumors (+58%) was added (previously missing) and values were written in red to be more visible.
In the Methods section, authors should provide subsections, including study design, inclusion and exclusion criteria, etc.
Response: Following this recommendation, we divided the Methods section into several sub-sections to address the study design, inclusion criteria, objectives and statistical analysis.
In respect to patients age (Table 1): 59 (16) and 58 (19). Authors indicated IQR, but they did not present Q1 and Q3 range only one value is provided: (16) or (19)?
Response: Data in Table 1 was modified accordingly – median age in cohort A 59 (IQR 16, Q1 51, Q3 67), in cohort B 58 (IQR 19, Q1 48, Q3 67).
Please, provide range for Ki67 (low) or (high) (Table 1).
Response: The range was specified in the table as Ki67 low (0-20%) and high (>20%). Subsequently, an editing error pertaining to the p-value of Ki67 expression in cohort A vs B was detected and corrected, as highlighted.
Also, in discussion section please add paragraphs related to clinical and practical aspects of the study.
Response: Following this recommendation, several practical clinical implications were derived and highlighted in lines 922-924 and 952-959 of the clean manuscript version.
How we can applicate your results into practice?, why your work is valuable in the field?
Response: We specified that our work should help Swiss oncologists acknowledge barriers in implementation on Oncotype RS testing (lines 226-231 of the clean manuscript version), as well as the necessity of inclusion of geriatric assessment (lines 913-915 of the clean manuscript version). Throughout our discussion we addressed the relevancy of genomic testing in therapy decision-making in order to establish if testing implementation is important in the daily clinic.
We also wanted to address where the Swiss landscape of genomic testing positions itself in regards to results from similar studies performed internationally.
Conclusions should be shortened and the authors should clearly present a summary of the research.
Response: The conclusion section underwent extensive editing. Results are now presented in a concise format.
Minor points:
1. Modification of the grammar and punctuation is required.
Response: Due to extensive editing, several grammar and punctuation changes were implemented. We hope we managed to address specific changes that were targeted.
Please provide strength of the study (Line 384).
Answers: Strengths of the study are now mentioned in lines 938 through 943 of the clean manuscript version.
There are several strengths of our study. Firstly, our institutions have extensive experience with Oncotype DX testing. Secondly, with over 320 patients, treated at a single oncological center divided between two hospitals, our study represents a relatively homogenous population. Thirdly, information retrieved from electronic database provided us with realistic information of what is actually happening in the daily clinical setting and at interdisciplinary tumor board meetings

Reviewer 2 Report
Comments and Suggestions for Authors
Review comment
This article titled as “Application of a 21-gene recurrence score in a Swiss single center breast cancer population. A comparative analysis of treatment administration before and after TAILORx” claimed that the aim of this study was to determine treatment patterns before and after publication of TAILORx at our Swiss breast cancer (BC) center with a focus on the intermediate RS nodal negative category. Authors conducted a retrospective study on 326 BC patients, of which 165 had a BC diagnosis before TAILORx (cohort A) and 161 after TAILORx publication (cohort B). Authors tried to observe how data derived from TAILORx analysis affected the daily clinical therapy decision-making process in a single oncology center from Basel, Switzerland, in order to determine if any significant changes in CHT application occurred especially in the intermediate RS, node negative BC patient cohort. Secondary endpoint was to determine leading therapy decision factors. They also looked at how tumor board (TB) decision deviated from standard recommendation as based on RS results. The changes we expected to observe were mainly based on lowering of RS thresholds and therefore on change in patient number. This study is well-designed, however, the results of this study failed to provide strong practical meaning for clinical practice. Significance of Content is low. Moreover, potential solutions to solve these problems raised by authors have also not been answered. Thus, major revision is recommended for current form. The main concerns have been listed as following:
1. as questions raised by authors “While this points to an undertreatment trend in our cohort, with all its implications – less toxicity for patients, less financial burden, higher risk - it raises questions about what is it that influences TB to decide against CHT-ET, when should they reconsider and what are the practical implications for patient survival and event free survival, for which a separate analysis of this cohort is highly recommended”, the potential answers should also be discussed in the article.
2.Please provide an explanation for the smiling face signal in line 97: “Ref😊”.
3. Based on the results of this study, the potential consequences should be thoroughly and comprehensively summarized and analyzed.
Author Response
Top of Form
Dear Ms. Peng,
Thank you for giving me the opportunity to submit a revised draft of my manuscript titled Application of a 21-gene recurrence score in a Swiss single center breast cancer population. A comparative analysis of treatment administration before and after TAILORx
to International Journal of Molecular Sciences.
We appreciate the time and effort that you and the reviewers have dedicated to providing your valuable feedback on my manuscript. We are grateful to the reviewers for their insightful comments on my paper.
We have been able to incorporate changes to reflect most of the suggestions provided by the reviewers. We have highlighted the changes within the manuscript. Here is a point-by-point response to the reviewers’ comments and concerns.
Comments from Reviewer 2
Review comment
This article titled as “Application of a 21-gene recurrence score in a Swiss single center breast cancer population. A comparative analysis of treatment administration before and after TAILORx” claimed that the aim of this study was to determine treatment patterns before and after publication of TAILORx at our Swiss breast cancer (BC) center with a focus on the intermediate RS nodal negative category. Authors conducted a retrospective study on 326 BC patients, of which 165 had a BC diagnosis before TAILORx (cohort A) and 161 after TAILORx publication (cohort B). Authors tried to observe how data derived from TAILORx analysis affected the daily clinical therapy decision-making process in a single oncology center from Basel, Switzerland, in order to determine if any significant changes in CHT application occurred especially in the intermediate RS, node negative BC patient cohort. Secondary endpoint was to determine leading therapy decision factors. They also looked at how tumor board (TB) decision deviated from standard recommendation as based on RS results. The changes we expected to observe were mainly based on lowering of RS thresholds and therefore on change in patient number.
This study is well-designed, however, the results of this study failed to provide strong practical meaning for clinical practice. Significance of Content is low. Moreover, potential solutions to solve these problems raised by authors have also not been answered. Thus, major revision is recommended for current form. The main concerns have been listed as following:
- as questions raised by authors “While this points to an undertreatment trend in our cohort, with all its implications – less toxicity for patients, less financial burden, higher risk - it raises questions about what is it that influences TB to decide against CHT-ET, when should they reconsider and what are the practical implications for patient survival and event free survival, for which a separate analysis of this cohort is highly recommended”, the potential answers should also be discussed in the article.
Response: After extensive editing, we provided some answers and specified possible future explorations in our discussion. As such, we mentioned that age and RS were relevant when deciding for or against chemotherapy, so we recommended implementing geriatric assessment (913-915 of the clean manuscript version). We also recommended acknowledging bias when selecting patients to undergo genomic testing and exploring this possibility through a prospective trial in which oncologists should decide before and after knowing RS results (939-942 of the clean manuscript version), as well as performing a sub-analysis analog RxPONDER trial to explore the role of menopausal status in the decision-making process (974-977 of the clean manuscript version).
2.Please provide an explanation for the smiling face signal in line 97: “Ref?”.
Response: Please excuse the error – it was not intended for the final version of the manuscript, was simply a note to myself to include the corresponding reference.
- Based on the results of this study, the potential consequences should be thoroughly and comprehensively summarized and analyzed.
Response: Following your recommendations, we re-wrote significant parts of the discussion and conclusion, as highlighted in the text.
Round 2
Reviewer 1 Report
Comments and Suggestions for Authors
The authors definitely implemented all my suggestions, so I suggest to publish the manuscript in its current form.
Reviewer 2 Report
Comments and Suggestions for Authors
This article provides certain novelty to clinical practice.